# Discovering Influential Positions in RFID-Based Indoor Tracking Data

**Ye Jin [1] and Lizhen Cui [1,2,*]**

[1]  School of Software, Shandong University, Jinan 250100, China; 201600150032@mail.sdu.edu.cn
[2]  National Engineering Laboratory for E-Commerce Technologies, Jinan 250100, China
*  Correspondence: clz@sdu.edu.cn

**Abstract:** The rapid development of indoor localization techniques such as Wi-Fi and RFID makes it possible to obtain users' position-tracking data in indoor space. Indoor position-tracking data, also known as indoor moving trajectories, offer many new opportunities to mine decision-making knowledge. In this paper, we study the detection of highly influential positions from indoor position-tracking data, e.g., to detect highly influential positions in a business center, or to detect the hottest shops in a shopping mall according to users' indoor position-tracking data. We first describe three baseline solutions to this problem, which are count-based, density-based, and duration-based algorithms. Then, motivated by the H-index for evaluating the influence of an author or a journal in academia, we propose a new algorithm called H-Count, which evaluates the influence of an indoor position similarly to the H-index. We further present an improvement of the H-Count by taking a filtering step to remove unqualified position-tracking records. This is based on the observation that many visits to a position such as a gate are meaningless for the detection of influential indoor positions. Finally, we simulate 100 moving objects in a real building deployed with 94 RFID readers over 30 days to generate 223,564 indoor moving trajectories, and conduct experiments to compare our proposed H-Count and H-Count* with three baseline algorithms. The results show that H-Count outperforms all baselines and H-Count* can further improve the F-measure of the H-Count by 113% on average.

**Keywords:** RFID; indoor space; indoor position-tracking data; indoor moving trajectory; influential position; H-count

## 1. Introduction

With the rapid development of indoor localization technologies [1], location-based services (LBS) and applications have been a new research topic in recent years [2]. Particularly, indoor localization devices make it possible to obtain people's moving trajectories in indoor space. Such indoor moving trajectories are useful for mining decision-making knowledge in many applications. For example, a museum can find the hottest visiting places based on the historical moving trajectories of users. A shopping mall can detect the most welcoming shops according to the tracking records of its customers. All these findings are helpful for the museum and shopping mall to adjust their layout of arts/products or to recommend products, leading to the improvement of service quality.

In this study, we aim to identify highly influential positions from indoor moving trajectories inside certain indoor spaces, like shopping malls, office buildings, and airports. A typical example is to find the most influential shops in a shopping mall. We notice that the position-tracking historical data of persons in a shopping mall usually represents people's interest in specific items, which is meant to detect highly influential positions, as, in many cases, people will not buy the same items (e.g., a refrigerator, or a wedding ring) again for a long time after they have bought one. Thus, it will be useful to tackle

people's indoor moving trajectories to know their potential buying interests. This is becoming a reality, due to the increasing deployment of indoor localization devices. For instance, most shopping malls now offer Wi-Fi services, which can be used to detect users' position in indoor space.

However, it is not a trivial task to detect highly influential positions from indoor moving trajectories [3]. The challenges are twofold. First, the positions in a specific indoor space are typically not visited by users equally. Taking a hotel as an example, the first floor usually has more visits than others, because all people have to visit the first floor when entering the hotel. In addition, some special positions, such as elevator rooms, are frequently visited. Thus, we cannot simply calculate the visiting number of indoor positions to detect influential positions. Second, the influence of an indoor position does not simply rely on the count of visits. The number of different users can reflect the covering range of users. Thus, it is necessary to consider both the visitor count and the number of users when detecting highly influential positions in indoor space.

Aiming to effectively detect highly influential positions from indoor moving trajectories, in this paper, we first formulate the problem and analyze three baseline solutions. Then, we propose a new algorithm, called H-Count. Furthermore, we present an improved version of H-Count, named H-Count*, which is demonstrated to be more effective than H-Count. Briefly, we make the following contributions in this paper:

(1) We formulate the problem of detecting highly influential positions from indoor moving trajectories and present three baseline solutions.
(2) Motivated by the H-index in the evaluation of authors/journals, we propose a new algorithm called H-Count for detecting highly influential indoor positions. H-Count adopts a similar idea to H-index and develops a mechanism to evaluate the influence of indoor positions. The greatest advantage of H-Count is that it does not only consider the visitor count but also considers the different visitors.
(3) We further present an improvement for H-Count, denoted H-Count*, by taking a filtering step to remove unqualified trajectories. This is based on the observation that a lot of visits to a location such as a gate are meaningless for the detection of influential indoor positions.
(4) We conducted experiments on trajectory data in a simulated indoor space with RFID readers and compared our H-Count and H-Count* with three baseline algorithms. The results in terms of precision, recall, and F-measure show that H-Count outperforms all baselines and H-Count* can further improve the performance of H-Count by 113% on average.

The remainder of the paper is structured as follows. In Section 2, we briefly describe the preliminaries and related work of this study. In Section 3, we analyze the baseline algorithms. In Section 4, we present the H-Count algorithm. Section 5 details the H-Count* algorithm. Section 6 reports the experimental results, and, finally, in Section 7, we conclude the paper.

## 2. Preliminaries and Related Work

### 2.1. Preliminaries

Indoor space has been studied for years [4–7]. In general, there are three kinds of indoor entities in indoor space, including indoor elements, sensor deployment, and moving object data. Figure 1 is the entity-relationship diagram of an indoor-space model [2]. The indoor element includes a room and a door, and the room is connected with the door. Moving objects are detected by sensors such as RFID readers; a sensor is deployed in the room or the door; a moving object is detected by the sensor when it moves in the indoor space.

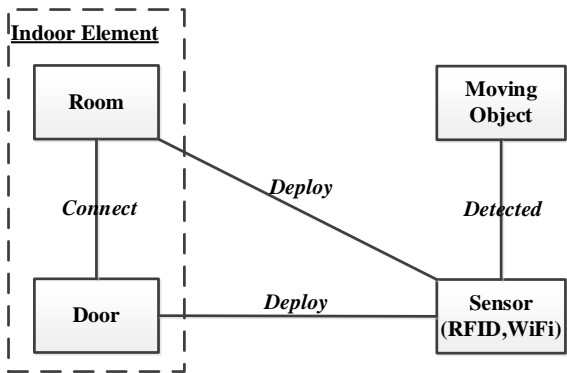

**Figure 1.** A simple model for indoor space.

In an indoor space, deploying RFID readers, we assume that every person in indoor space has an identity card embedded with an RFID tag. This is very common in current office buildings, where every person must have a pass card to enter into the building. As RFID tags are much cheaper, it will not cost too much to offer all employees pass cards with RFID tags. As all RFID readers have been deployed in the indoor space in advance, we can know the position of each RFID reader. When a person with an RFID tag passes by an RFID reader, the reader will generate a sensing record including the RFID tag ID and time, which will be stored in a centralized database. In this way, we can track the movement of all persons in indoor space and obtain indoor position-tracking data over a period.

Generally, indoor position-tracking data, also known as indoor moving trajectories, tracks a series of time-stamped indoor positions, which can be defined as follows:

Here, *mo* is the moving object and $p_i$ is an indoor position represented by an RFID reader's position in indoor space.

$$TR_{mo} = \{(mo, \langle p_1, p_2, \ldots, p_n, \rangle) \big| (\forall p_i)\ (p_i \in LOC_{mo} \wedge (p_i.t_e < p_{i+1}.t_s))\} \tag{1}$$

## 2.2. RFID-Based Indoor Positioning

Radio frequency identification (RFID) technology is a kind of wireless identification technology that can identify the target without human interference. Through the microchip label of the mobile object, the information can be connected to the computer network to identify, track, and confirm the status of the mobile object. Each RFID tag has a unique ID code, which can provide sufficient object information. An RFID system [8] is mainly composed of RFID readers, RFID tags, antennas, computers (PC or MCU), etc. Among these, the RFID reader is also called an RFID query device. It can not only transmit certain radio carrier signals through the antenna but also receive certain information. RFID readers are generally deployed in key indoor areas, such as doors, corridors, elevators, and stairs. They can sense and read RFID tag information. RFID tags are divided into passive tags, active tags, and semi-passive (also known as semi-active) tags. There is no power supply inside the passive tag, and the internal integrated circuit is driven by electromagnetic signals from the RFID readers. When a signal of sufficient strength is received, the RFID tag can send data to an RFID reader, mainly including an ID number (tag unique number) and information stored in EEPROM (electrically erasable programmable read-only memory) in the tag in advance. The specifications of the semi-passive label are similar to those of the passive label, except that a small battery can drive the IC in the label. If the IC only receives a weak signal from the reader, the label still has enough power to send the memory data back to the reader. Generally speaking, unlike passive and semi-passive, an active tag contains a power supply itself, which provides the power required by the internal IC to generate external signals. It has a long reading distance, can accommodate a large memory capacity, and can also store some additional information from the reader. Active tags can transfer memory data to readers and writers at any time

with the help of internal power [9]. Due to the low price and small size of passive RFID tags, there is no need for a power lamp. At present, the RFID tags used in the market are mainly passive electronic tags.

When a mobile object enters a room with an RFID tag, the tag stores a small amount of information on the mobile object. Each RFID reader transmits the signal outward through the antenna. When a mobile object enters the coverage of an RFID reader, the RFID tag carried by the mobile object will be able to receive the signal sent by the reader and then transmit the information through the antenna. In this way, it will be received by the RFID reader, and then the reader will demodulate and decode the received signal, send useful information to the server through a wired or wireless communication link, and record the position of the object. Due to the limited coverage of RFID readers, RFID cannot cover the whole indoor space when deployed. Therefore, in the area where we cannot collect the records of mobile objects, we can use some estimation methods to estimate the position of mobile objects. RFID can also identify high-speed moving objects and detect multiple tags quickly. Therefore, RFID technology has been widely used in logistics and supply management, product manufacturing and assembly, animal identification and tracking, security inspection, and other fields.

### 2.3. Influential Indoor Position Discovery

Discovering influential indoor positions has been a promising area in the analysis of indoor position-tracking data [10]. However, existing works are mainly focused on outdoor space and few of them are toward indoor position-tracking data.

Influential positions, which are called important positions or hot positions in some of the literature, have been studied in outdoor space for years. Alvares et al. [11] proposed a method of analyzing moving trajectories in a road network, which only considered the geometric characteristics of trajectories. They presented definitions of stops and moves to identify the important points in indoor moving trajectories and further proposed a density-based algorithm to detect important positions. However, stops are limited to pre-determined areas, such as intersections with traffic lights, hotels, and scenic spots, which are difficult to know in advance in real environments. The idea of stop location inspired a number of subsequent studies [12,13], which mainly used the stop locations to find out the hot spots or areas in users' moving trajectories. The hot spots or areas can be used to measure the similarity of users' travel, to extract Top-k interesting locations, etc. All these works concerned outdoor space and were not suitable for indoor space.

Concerning indoor space, Hussein et al. [14] proposed a graph-based indoor model and presented an algorithm for detecting hot locations in indoor space. This algorithm considered the number of users accessing an indoor location as well as the users' stay time in the location. Such an idea is based on visitor count and visiting time duration, which are regarded as baseline algorithms in this study and we will discuss them in Section 3. This algorithm will also be compared in the experiments. However, the visiting-count-based idea cannot correctly reflect the impact of an indoor location because some special locations such as the lobby, corridors, and lift rooms usually have higher visiting frequencies than others.

Ahmed et al. [15,16] proposed a density-based algorithm to detect highly influential locations based on indoor moving data. Their study was especially about the characteristics of airport parcel transport devices, which are much different from those of people's movements in indoor space. For example, while airport parcels are transported in a fixed direction at a steady speed, people may move in a shopping mall with different speeds and directions. To this end, their algorithm cannot be regarded as a general solution to the discovery of highly influential indoor locations. The density-based idea was also employed in recent work [17], which aimed to detect dense regions but not positions in indoor space.

Jin et al. [18] proposed a new algorithm for discovering indoor hotspots based on mutual reinforcement between indoor locations and users. They assume that users have different interests in different locations, and each visit of a user to a location represents voting for the location, which will lead to an increase in the location's score. Meanwhile, if a user visits a high-scoring location, the user's

score will also be added. However, this approach needs to know the interests of all users in each location, which is not practical in real applications. For example, it is not possible to know each customer's interest in all shops in a shopping mall in advance. Thus, in our experiment, we will not compare our proposal with this algorithm, but only consider other solutions, including visiting-count-based, visiting-duration-based, and density-based algorithms. These baseline algorithms will be briefly introduced in Section 3.

## 3. Baseline Algorithms

In this section, we discuss three baseline algorithms for detecting influential positions from indoor tracking data. The first is a counting-based algorithm (see Section 3.1), the second is a density-based algorithm (see Section 3.2), and the third is a duration-based algorithm (see Section 3.3).

### 3.1. Count

The first baseline algorithm detects indoor influential positions based on the count of visits [14]. Thus, the position with the highest visitor count will be regarded as the most influential position. The counting algorithm process is shown in Algorithm 1.

---
**Algorithm 1:** Count.

---
1:　　**Input**: $ST = \{ST_1, ST_2, \cdots, ST_m\}$, indoor trajectories
2:　　**Output**: $H = \{H_1, H_2, \ldots, H_n\}$, vector of hotness of all indoor positions
3:　　　**for** $i = 0$ to $n$ **do** $H_i \leftarrow 0$; *//initializing H*
4:　　　**for** $i = 0$ to m **do** $H_{ST_i.R\_ID} \leftarrow H_{St_i.R\_ID} + 1$; *//$H_{St_i.R\_ID}$ is the visitor count of room R_ID*
5:　　　**return** $H$

---

### 3.2. Density

The density of a position refers to the visitor count during a time unit [15–17]. This metric is different from the simple count, as we have discussed in Section 3.1. For example, perhaps two positions both have 100 visits, and position A receives 100 visits within 10 min while position B receives 100 visits within 20 min. In such a case, we can know that the density at position A, which is 10 visits/min, is higher than that at position B, which is 5 visits/min.

According to this idea, we present the density algorithm in Algorithm 2. In this algorithm, we first calculate the visiting time of each position, which means the time interval between the first visiting time and the last visiting time to the position. We then calculate the total visits to each position. Finally, we get the density for each position.

---
**Algorithm 2:** Density

---
1:　　**Input**: $ST = \{ST_1, ST_2, \cdots, ST_m\}$, indoor trajectories
2:　　**Output**: $H = \{H_1, H_2, \ldots, H_n\}$, vector of hotness of all indoor positions.
3:　　**Preliminary**: $LT = \{LT_1, LT_2, \ldots, LT_p\}$, $LT_i$ is the set of the tracking records of position $i$.
4:　　　**for** $i = 0$ to $n$ **do** $H_i \leftarrow 0$; *//initializing H*
5:　　　**for** $i = 0$ to p **do** $D_i \leftarrow 0$; *//initializing D*
6:　　　**for** $i = 0$ to $n$ **do**
7:　　　　**for** $j = 0$ to m **do**
8:　　　　　**if** $ST_j.R\_ID == i$ **do**
9:　　　　　　put $ST_j$ *into* $LT_i$;*//prepare the trajectories for position i*
10:　　　**for** $i = 0$ to $n$ **do**
11:　　　　$visits \leftarrow visiting\_count(LT_i)$;
12:　　　　$time \leftarrow visiting\_time\_span(LT_i)$;
13:　　　　$H_i \leftarrow (visits / time)$; *//get the density for position i.*
14:　　　**return** $H$

---

*3.3. Duration*

The above two baseline algorithms only concern the visitor count of each position but neglect the visiting time duration. Thus, in this algorithm, we consider the period of each visit to a position. The basic idea is that, if people stay at a position for a long time, this position is probably highly influential because the staying time can reflect people's interest in the position.

The duration algorithm is shown in Algorithm 3.

---

**Algorithm 3:** Duration

---

1:    **Input**: $ST = \{ST_1, ST_2, \cdots, ST_m\}$, indoor trajectories
2:    **Output**: $H = \{H_1, H_2, \ldots, H_n\}$, vector of hotness of all indoor position
3:       **for** $i = 0$ to $n$ **do** $H_i \leftarrow 0$; *//initializing H*
4:       **for** $i = 0$ to $n$ **do**
5:         **for** $j = 0$ to m **do**
6:           **if** $ST_j.R\_ID == i$ **do**
7:             $H_i \leftarrow H_i + (ST_i.LeaveTime - ST_i.EnterTime)$ ;
8:       **return** $H$

---

## 4. H-Count: An H-Index-Based Approach

In this section, we present a new algorithm for detecting highly influential positions from indoor trajectories, which is called H-Count. This algorithm is motivated by the H-index [19], which has been widely adopted in academia to evaluate the impact of a researcher or a venue of publication. The idea of the H-index is to calculate the maximum value of $h$ such that the given author/journal has published $h$ papers that have each been cited at least $h$ times. The H-index is designed to improve upon simpler measures such as the total number of citations or publications.

Regarding the problem of this study, we found that the task of finding out the indoor influential positions is similar to the evaluation of authors/journals, i.e., to find out influential authors/journals. Thus, we propose to map the problem of detecting influential indoor positions as a similar problem to calculate the H-index of each indoor position. Hence, we can calculate the maximum value of $h$ for a given position such that the position has been visited by at least $h$ users and each user has at least $h$ visits to the position.

Algorithm 4 shows the H-index-based approach, which is called H-Count in this paper. To calculate the H-Count for each position, we first summarize the visitor count of each user to each position and then calculate the H-Count using the prefix-sum method.

---

**Algorithm 4:** H-Count

---

1:    **Input**: $ST = \{ST_1, ST_2, \cdots, ST_m\}$, indoor trajectories
2:    **Output**: $H = \{H_1, H_2, \ldots, H_n\}$, vector of hotness of all indoor positions.
3:    **Preliminary**: $p$ is the number of visitors. $C = \{C_1, C_2, \ldots, C_p\}$ is the vector of visitor count. $P = \{P_1, P_2, \ldots\}$ is the vector of the prefix sum of H-Count.
4:       **for** $i = 0$ to $n$ **do** $H_i \leftarrow 0$; *//initializing H*
5:       **for** $i = 0$ to p **do** $C_i \leftarrow 0$; *//initializing C*
6:       **for** $i = 0$ to $n$ **do**
7:         **for** $j = 0$ to m **do**
8:           **if** $ST_j.R\_ID == i$ **do**
9:             $C_{ST_j.M\_ID} \leftarrow C_{ST_j.M\_ID} + 1$; *//calculating count, M_ID is the visitor id*
10:         **for** $j = 0$ to max{C} **do** $P_j \leftarrow 0$; *//initializing P*
11:             $k \leftarrow \max\{n | P_n \neq 0\}$;
12:       **repeat***//calculating H-count using prefix sum*
13:             $k \leftarrow k - 1$;
14:             $H_i \leftarrow H_i + P_k$;
15:       **until** $k == 0$ or $H_i \geq k$
16:             $H_i \leftarrow k$;
17:       **return** $H$

---

## 5. H-Count*: Further Improvement of H-Count

In this section, we further present an improvement of the H-Count algorithm. We noticed that some special positions in indoor space may have an extraordinary visitor count but such a high visitor count cannot reflect the interests of users. For example, the gate of a building will always get almost the highest visitor count according to indoor tracking records. However, it does not mean that the gate is the most interesting or influential position in indoor space. There are a few similar positions in a building, such as stair rooms, elevator rooms, pathways, and gates. The tracking records of such special positions will probably impact the precision of the detection of highly influential positions in indoor space.

To tackle this problem, we propose a filtering approach to remove those irrelevant tracking records from the original data set. We have found that most visits to positions like a gate have a very short staying time. This is mostly because users just pass by the position. Therefore, we first define the normal indoor position for a moving object as follows.

*Definition 5 (Normal Indoor Position).* A normal indoor position $NP_{mo}$ of an indoor moving object *mo* is defined as follows:

$$NP_{mo} = \{p.s | p \in LOC_{mo} \wedge (p.t_e - p.t_s > \varphi)\} \tag{2}$$

where $\varphi$ is a pre-defined time threshold.

Definition 5 shows that only those tracking records having a time duration longer than a given threshold will be regarded as normal indoor positions.

Next, we define the normal indoor trajectory.

*Definition 6 (Normal Indoor Trajectory).* A normal indoor trajectory $NT_{mo}$ of an indoor moving object *mo* is defined as a trajectory that only travels the normal indoor locations of *mo*:

$$NT_{mo} = \{\langle p_1, p_2, \ldots, p_n \rangle | (\forall p_i)(p_i \in NP_{mo})\} \tag{3}$$

Generally, the normal indoor trajectory of a user refers to tracking records that only record the historical movement passing by normal indoor locations. As we have introduced a staying threshold for each normal indoor location, if a user simply passes by a location (e.g., a gate), the tracking record will not remain in the normal indoor trajectory of the user. With such a design, we can avoid the influence of special tracking records on the effectiveness of detecting highly influential locations in indoor space.

---

**Algorithm 5:** *Filter*

---

1:　**Input**: *TR* : the original trajectory set; *MO*: the set of moving objects; $\varphi$.
2:　**Output**: *ST*: the set of normal trajectories
3:　　$ST \leftarrow \varnothing$;
4:　　　**for** each $obj \in MO$ **do**
5:　　　　**for** each $tr \in TR$ **do**
6:　　　　　**if** $tr.mo = obj$　　**then**//*tr is a trajectory of the current obj*
7:　　　　　　　$temp \leftarrow normal\_location(tr, \varphi)$; //*save the normal locations in tr*
8:　　　　　　　//*merge the normal locations into the current normal trajectory*
9:　　　　　　　$ST_{obj} \leftarrow merge(ST_{obj}, temp)$;
10:　　　　**end if**
11:　　　**end for**
12:　　$ST \leftarrow ST \cup ST_{obj}$ ;//*put the updated trajectory into the trajectory set*
13:　**end for**
14:　**return** *ST*

---

Algorithm 5 shows the extraction of the normal indoor trajectories from the original tracking records. The sub-routine *normal_location* (*tr*, $\varphi$) returns the positions that the moving object stays

in over the time threshold $\varphi$ in the trajectory *tr*, as well as the stay time. Then, we extract the positions in the list *temp* as the normal indoor positions for the object *obj*.

Based on the *filter* algorithm, we can present the improved H-Count algorithm, which is called H-Count*, as shown in Algorithm 6.

---

**Algorithm 6:** H-*Count*\*

---

1:    **Input**: $ST = \{ST_1, ST_2, \cdots, ST_m\}$, indoor trajectories
2:    **Output**: $H = \{H_1, H_2, \ldots, H_n\}$, vector of hotness of all indoor positions.
3:    $NT \leftarrow filter(ST)$ ;//*filtering*
4:    $H \leftarrow$ H-*Count*$(NT)$;
5:    **return** $H$

---

## 6. Performance Evaluation

In this section, we report the experimental results. Section 6.1 details the experimental setting, including the data set, metrics, configurations, and compared methods. In Section 6.2, we present detailed results and analysis.

### *6.1. Setting*

#### 6.1.1. Data Set

We simulated the building of the school of software at our university and generated indoor tracking data using an indoor data generator called *IndoorSTG* [20]. *IndoorSTG* can simulate different indoor spaces consisting of various elements, including rooms, doors, corridors, stairs, elevators, and virtual positioning devices such as RFID or Bluetooth readers. The simulated building has six floors, and there a total of 94 RFID readers were deployed to represent different types of indoor elements. We simulated 100 moving objects in such an indoor space for 30 days, and, finally, generated 223,564 position-tracking records. These records were then used as the experimental data. Table 1 shows an example of indoor position-tracking data generated by *IndoorSTG*.

**Table 1.** An example of indoor tracking records.

| Reader_ID | Object_ID | Enter_Time | Leave_Time |
|:---:|:---:|:---:|:---:|
| 4 | 1 | 2014-03-08 07:48:00 | 2014-03-08 07:59:15 |
| 9 | 2 | 2014-03-08 08:10:32 | 2014-03-08 08:10:36 |
| 15 | 3 | 2014-03-08-08:32:12 | 2014-03-08-08:55:22 |

#### 6.1.2. Metrics

We mainly focused on the measurement of the effectiveness of influential indoor position detection. Thus, we mainly compared our proposal with existing algorithms in terms of precision, recall, and F-measure, which are all general metrics to evaluate the performance of information extraction and knowledge discovery methods [21]. Note that the time performance is not a crucial metric, because influential indoor position detection is not an online job and can be executed as a background program in the server storing trajectories. For instance, a shopping mall can run the program every weekend to find the influential indoor positions during the past week. Therefore, in our experiments, we will not present the results of time performance. To measure the effectiveness of our proposal, we first asked 10 students in our university, who were very familiar with the building of the school of software, to manually annotate each position, using the following scores shown in Table 2. The positions with an average score of over 1.5 were annotated as influential indoor positions.

**Table 2.** Users' rating of an indoor position.

| Ratings | Description |
|---------|-------------|
| 2 | *Influential* |
| 1 | *Probably influential* |
| 0 | *Neutral* |
| −1 | *Probably not influential* |
| −2 | *not influential* |

### 6.1.3. Configuration

All of the algorithms were implemented in Java on Windows 7 and were run on a PC with an Intel(R) Core(TM) i3-3220 CPU @3.30GHz and 4GB DDR2 memory. The time threshold φ for detecting stay trajectories is set to 10 min by default. The influence of φ on the performance will be measured in the following experiments.

### 6.1.4. Compared Methods

All of the experiments were performed on a computer with an Intel Core i7-7700 3.6 GHz CPU, 16 GB DRAM, and a 512 GB SSD. All algorithms were implemented in Java on Windows 10. The time threshold φ for detecting normal indoor positions and trajectories was set to 10 min by default. The influence of φ on the performance was measured in the experiments.

### 6.2. Results

### 6.2.1. Effectiveness of H-Count

Figures 2–4 show the precision, recall, and F-measure of our proposed H-Count and three other baseline algorithms, namely Count, Density, and Duration. All three metrics show that our H-Count algorithm achieves the best performance when measuring the top 5 to 40 influential indoor positions, indicating that both visitor count and the different numbers of visitors are important to the identification of influential indoor positions. We did not compare numbers over 40 because there are only 94 positions in the data set, and it is meaningless to return a majority of indoor positions. For example, if we verify the performance of each algorithm over all 94 positions, i.e., verifying the top 94 performance, all the metrics will be 100%. Meanwhile, in real indoor spaces such as shopping malls, people only need to know the most influential indoor positions, e.g., the hottest shops during the past month, based on which they can take necessary decisions such as changing the layout of products.

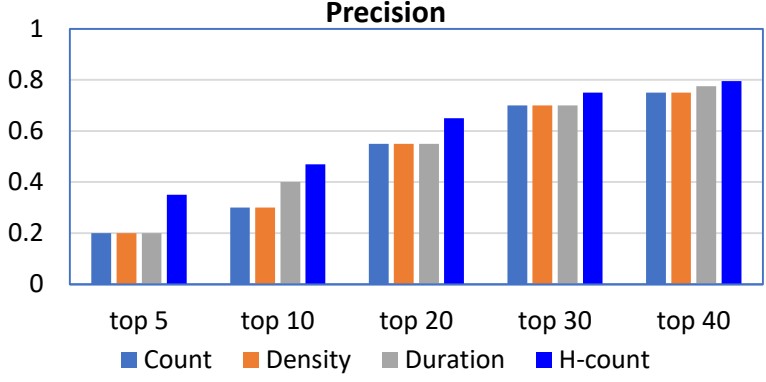

**Figure 2.** Precision comparison between H-Count and three baselines.

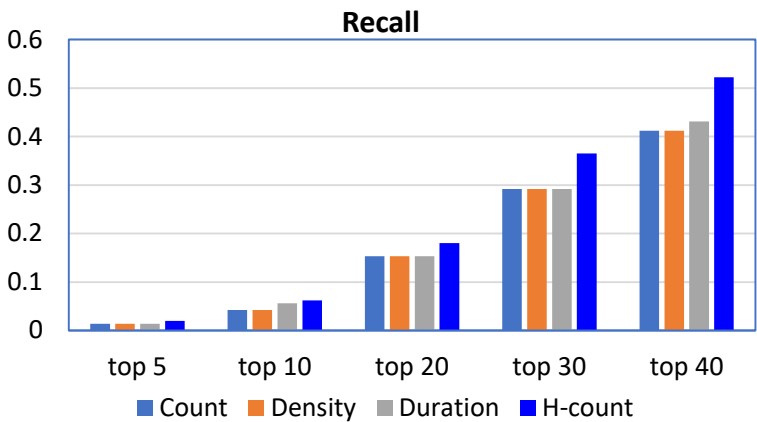

**Figure 3.** Recall comparison between H-Count and three baselines.

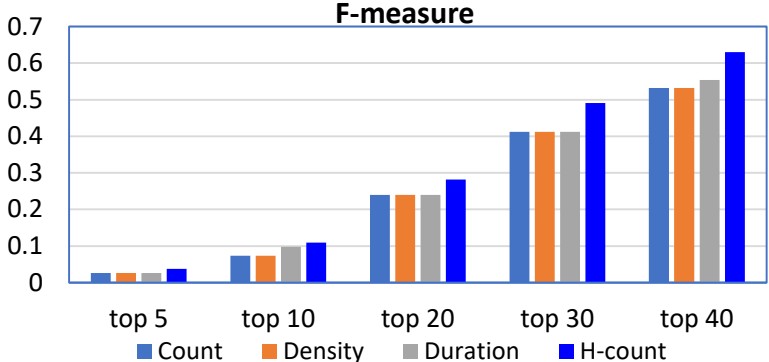

**Figure 4.** F-measure comparison between H-Count and three baselines.

As Figures 2 and 3 show, although H-Count achieves the best precision and recall among all the settings, the overall precision and recall of the algorithms was much lower than we expected. This is mainly because the experimental dataset tracks all the positions of people in a given period. Note that, in addition to the office rooms in the indoor space, we also deployed RFID readers in elevators, rooms, stair entrances, gates, corridors, etc. Compared to office rooms, we found that those positions had much higher visitor counts. This is easy to understand, as people have to pass through the gate and corridors whenever they enter into the indoor space. In our experiment, each moving trajectory starts at the gate and ends when the moving object arrives at the targeted room. Therefore, we can infer that all gates, corridors, elevator rooms, and stair rooms will have more visits than normal office rooms. However, this is very common in real indoor environments such as office buildings and shopping malls. As a result, if we do not do any preprocessing on the raw dataset, all algorithms will recognize positions like gates and corridors as influential or hot positions, which are not meaningful for real applications. To this end, a preprocessing step is necessary to remove the unqualified indoor tracking data, which is the basic idea of the H-Count* algorithm.

6.2.2. H-Count vs. Filter-based H-Count*

Next, we verified the performance of the filtering approach that we adopted in H-Count*. Figures 5–7 show the comparison of precision, recall, and F-measure between H-Count and H-Count*. As Figure 5 shows, H-Count* achieved much higher precision than H-Count, due to its filtering strategy. However, as shown in Fig. 6, the recall of both algorithms was quite low. This is because the recall metric depends on the number of returned results. Thus, when we only measure the top 5 or 10 results, the recall value is definitely quite low, because at most five or ten relevant results can be returned. To this end, the precision and F-measure metrics are more meaningful. As Figure 7 shows, the F-measure of H-Count* improved by 113% on average. Especially, when returning top

5, 10, and 20 positions, compared with H-Count, H-Count* improves the F-measure by 220%, 98%, and 47%, respectively. This result is particularly useful in real applications, as many applications need to detect the most influential indoor positions, e.g., to detect the top 10 influential indoor positions in an airport building. To this end, we can say that the filter-based H-Count* is more suitable for indoor-space applications. The main reason for the high performance of H-Count* is that it removed the most irrelevant indoor positions in trajectories. This can enhance the quality of indoor moving trajectories, yielding a performance improvement in detecting influential indoor positions.

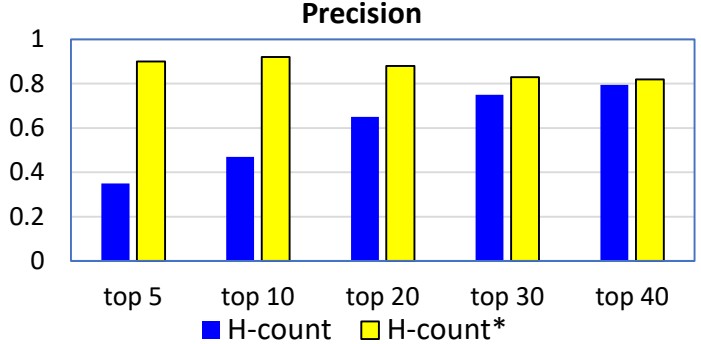

**Figure 5.** Precision comparison between H-Count and H-Count*.

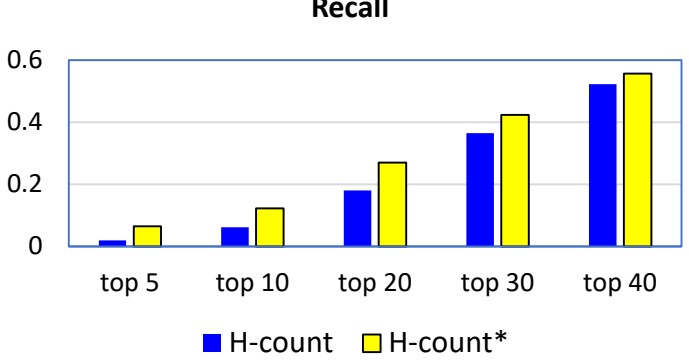

**Figure 6.** Recall comparison between H-Count and H-Count*.

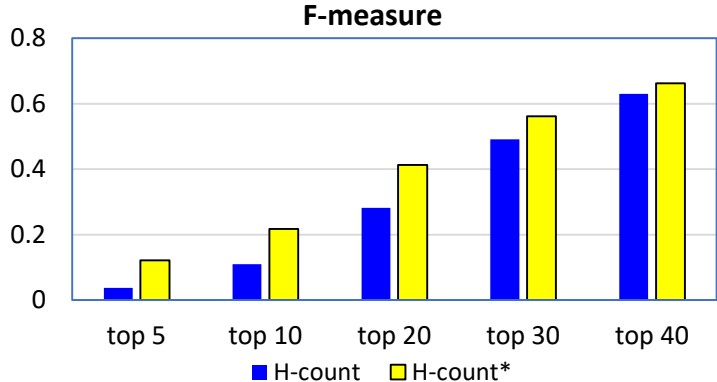

**Figure 7.** F-measure comparison between H-Count and H-Count*.

### 6.2.3. H-Count* vs. Filter-Based Baselines

To measure the effectiveness of the filtering strategy on other methods, in this experiment, we implemented filter-based baseline algorithms including Count, Density, and Duration. We denote them as Count*, Density*, and Duration*, respectively. Here, Count* refers to the filter-based Count algorithm, in which we first invoke the Filter algorithm to generate the normal indoor trajectories.

Figure 8 shows the comparison of H-Count* and the other three baselines concerning F-measure. We can see that the filtering strategy is also helpful to improve the performance of the baseline algorithms. All of them have achieved a better F-measure compared with the implementations not invoking the Filter algorithm. In addition, H-Count* also outperforms all the baseline algorithms.

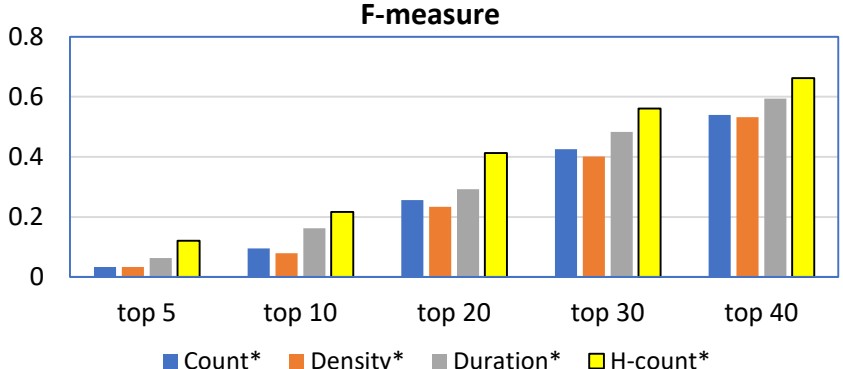

**Figure 8.** F-measure comparison between H-Count* and three filter-based baselines.

6.2.4. Impact of Parameter

φ. In this experiment, we measured the impact of the parameter φ, which refers to the time threshold that is used to filter indoor trajectories. This parameter is designed to reflect the users' interest in indoor positions. Thus, we empirically set it to 10 min by default in all previous experiments. This is because we believe a 10-min stay at a specific position is somehow enough to reflect the users' interest in the position. However, to reveal the impact of other possible values on the parameter, we set different values and checked the change in the precision of H-Count*. The results are shown in Figure 9, where we can see that when the value is set to 10 min, H-Count* achieves the highest precision. Generally, the setting of the parameter depends on the properties of the trajectories. For example, in a museum, it is probably necessary to set a value higher than 10 min because people tend to spend more time visiting the arts in a museum. In real applications, we can run a small set of training data to find out the appropriate value of the parameter in advance.

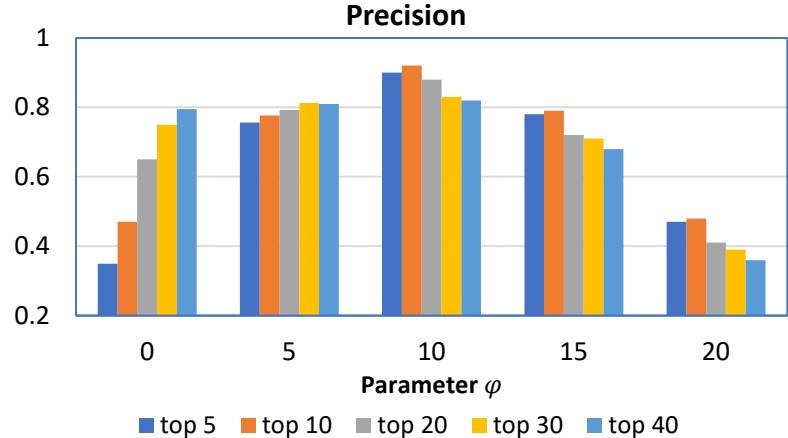

**Figure 9.** Impact of the parameter φ on the precision of H-Count*.

## 7. Conclusions

With the rapid development of the Internet of Things (IoT) and indoor positioning technologies such as Wi-Fi and RFID, indoor moving objects have become a new research hotspot. Based on the unique features of indoor space and the urgent need for indoor location-based services, in this paper, we studied the detection of highly influential indoor positions from indoor tracking records. We first

formulated the problem and then analyzed three baseline solutions. Next, motivated by the H-index in the evaluation of authors/journals, we proposed a new algorithm called H-Count. H-Count adopts a similar idea to the H-index and develops a mechanism to evaluate the influence of indoor positions. The greatest advantage of H-Count is that it does not only consider the visitor count but also considers the different visitors. We further presented an improvement for H-Count by taking a filtering step to remove unqualified trajectories before performing H-Count. This is based on the observation that a lot of visits to a position such as a gate are meaningless for the detection of influential indoor positions. We conducted experiments on a simulated indoor space and simulated trajectories and compared our proposed H-Count and H-Count* with three baseline algorithms. The results show that H-Count outperforms all baselines and H-Count* can further improve the F-measure of H-Count by 113% on average.

Although the experiment in this study was based on people's moving trajectories in RFID-enabled indoor space, the proposed approach can also be applied to trajectories generated by other indoor positioning technologies, such as Wi-Fi and Bluetooth. This is because we only track the stopping locations of users but do not care about how the position information is detected in indoor space. So far, compared to Wi-Fi and Bluetooth, RFID technology can precisely capture people's locations in indoor space by deploying RFID readers in indoor space. However, this is not the only choice. With the improvement in the positioning precision of other indoor positioning technologies, especially the Wi-Fi-based technology [22] that has been deployed in most indoor spaces, we can expect that, shortly, most buildings will provide Wi-Fi-based indoor positioning support that can precisely detect people's locations in indoor space.

Another issue that is orthogonal to this study is how to preserve people's privacy in indoor location-based services [23]. This privacy issue has been widely studied in trajectory data analysis [24] and location-based services [25] because many people, especially those from European Union countries, have an urgent need to preserve privacy. Some laws have also been issued by European countries. Presently, there are several ways to preserve privacy when capturing people's moving trajectories in indoor space, e.g., by using an encryption approach [26] or a privacy-preserving protocol [27]. Generally, most privacy-preserving approaches designed for outdoor-space-based location services can be slightly modified to suit indoor-space applications.

In the future, we will investigate other possible algorithms to detect influential locations from indoor tracking data. One direction is to employ a machine learning method, e.g., the reinforcement learning model [28]. Based on the reinforcement learning model, one possible solution is to construct a model of users and positions and develop a rewarding mechanism to dynamically update the scores of each position. However, how to construct the reinforcement learning framework and how to define the reward mechanism are challenging issues.

**Author Contributions:** Y.J., conceptualization, data curation, methodology, validation, writing—original draft preparation; L.C., funding acquisition, project administration, supervision, and writing—review and editing. All authors have read and agreed to the published version of the manuscript.

**Funding:** This research was funded by the National Science Foundation of China (grant number: 91846205).

**Acknowledgments:** We would like to thank the editors and anonymous reviewers for their suggestions and comments to improve the quality of the paper.

**Conflicts of Interest:** The authors declare no conflict of interest. The funders had no role in the design of the study; in the collection, analyses, or interpretation of data; in the writing of the manuscript; or in the decision to publish the results.

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
