# Peer review of "Discovering Influential Positions in RFID-Based Indoor Tracking Data"

_information, doi:10.3390/info11060330_

Round 1

Reviewer 1 Report

The structure of the paper is very clear. From state of the art till algorithm description till dissuasion, the paper is well organized and written. However my only issue with the paper is its novelty. To me the proposed algorithm is just a simple modified version of the existing one. The modification is also straight forward with only one time constraints. I am not sure whether the novelty is sufficient to grant a journal publication. Please either emphasize the novelty a bit deeper or add additional novelty to the paper in the revised version.  

Author Response

Response to Referee#1 ______________________________

Referee#1’s major concerns about this paper.

The structure of the paper is very clear. From state of the art till algorithm description till dissuasion, the paper is well organized and written. However, my only issue with the paper is its novelty. To me, the proposed algorithm is just a simple modified version of the existing one. The modification is also straight forward with only one time constraints. I am not sure whether the novelty is sufficient to grant a journal publication. Please either emphasize the novelty a bit deeper or add additional novelty to the paper in the revised version. 

RESPONSE: Thanks for your comments. The novelty and new contributions of this article are twofold. First, we propose a new algorithm called H-Count for detecting influential positions from RFID-based indoor tracking data. This algorithm is inspired by the H-index design but we apply the H-index idea into the research problem of influential position detection. We believe that the H-Count algorithm is a new design that can be valuable for other researches in this field. Second, we present an improvement of the H-Count algorithm by filtering trajectories with low quality. This is based on our observation of people’s moving properties in real indoor space. Although the filtering idea is simple, we emphasize that this improvement can achieve much better performance than H-Count. We also add a new paragraph in the conclusion part discussing another novel idea, namely the reinforcement-learning-based algorithm, to solve the same problem of this study, but we will investigate the details in the future.

Reviewer 2 Report

This paper presents a new metric to measure the influential position in the indoor environment. Although the metric is inspired by the H-index which measures the influence of an author in academia, its applicability is well justified in the indoor positioning scenario. In addition, the performance of the new metric is well evaluated through data collected in the real environment and compared with other general metrics like F measure in information extraction and knowledge discovery. Overall it is a solid work.

The scenario is reasonable and well justified. The design is clear and easy to follow. The only concern is performance evaluation. 

  • In Fig. 2 and Fig. 3, the precision and recall are quite low for the top 5 and top 10. It would be better if the authors can provide some explanation about the phenomenon. 
  • In Fig. 7, it shows when \PHI is set to be 10mins, the precision can be as high as 80% even for the top 5 influential positions. It is not consistent with previous results. The authors are expected to justify the result.

Some typo but not affect understanding

- identifying card -> identity/identification card

Author Response

Response to Referee#2 ______________________________

Referee#2’s major concerns about this paper.

1. In Fig. 2 and Fig. 3, the precision and recall are quite low for the top 5 and top 10. It would be better if the authors can provide some explanation about the phenomenon.

RESPONSE: Thanks for your comments. We have added a new paragraph before Fig. 2 to explain why the precision and recall of all the algorithms is quite low. As Figs. 2 and 3 show, although H-Count achieves the best precision and recall among all the settings, the overall precision and recall of all the algorithms is much lower than we expect. This is mainly because the experimental dataset tracks all the positions of people in a given period. Note that in addition to the office rooms in the indoor space, we also deploy RFID readers in elevators rooms, stair entrances, gates, corridors, etc. Compared to office rooms, we found that those positions have much higher visiting counts. This is easy to understand, as people have to pass through the gate and corridors whenever they enter into the indoor space. In our experiment, each moving trajectory starts at the gate and ends when the moving object arrives at the targeted room. Therefore, we can infer that all gates, corridors, elevator rooms, and stair rooms will have more visits than normal office rooms. However, this is very common in real indoor environments such as office buildings and shopping malls. As a result, if we do not make any preprocessing on the raw dataset, all algorithms will recognize positions like gates and corridors as influential or hot positions, which are not meaningful to real applications. To this end, a preprocessing step is necessary to remove the unqualified indoor tracking data, which is the basic idea of the H-Count* algorithm.

2. In Fig. 7, it shows when \PHI is set to be 10mins, the precision can be as high as 80% even for the top 5 influential positions. It is not consistent with previous results. The authors are expected to justify the result.

RESPONSE: Thanks for your comments. To explain the result in Fig. 7 more clearly, we have included the precision and recall compassion between H-Count and H-Count* in the revised paper, where the new Fig. 5 shows precision and Fig. 6 shows recall. As Fig. 5 shows, H-Count* achieves much higher precision than H-Count, due to its filtering strategy. Especially when the threshold \PHI is set to 10 minutes, the precision of H-Count* reaches 0.9. Another point is that the recall of both algorithms is quite low. This is because the recall metric depends on the number of returned results. Thus, when we only measure the top 5 or 10 results, the recall value is definitely quite low, because at most five or ten relevant results can be returned. As a result, we have added new figures (Figs. 5 and 6) in the revision and updated the discussions.

3. Some typo but not affect understanding

- identifying card -> identity/identification card

RESPONSE: Thanks for your comments. We have carefully proof-read the entire paper and corrected typos and some writing mistakes.

Reviewer 3 Report

This paper explored the detection of highly-influential positions from indoor position-tracking data. It proposed a new algorithm called H-Count, which evaluates the influence of an indoor position. The work simulates 100 moving objects in a real building deployed with 94 RFID readers during 30 days to generate 223,564 indoor moving trajectories, and conduct experiments to compare the H-Count and H-Count* with three baseline algorithms. The results show that H-Count outperforms all baselines and H-Count* can further improve the F-measure of H-Count by 113% on average.

This paper gives a clear description of the issue to be addressed, and which is an interesting topic indeed. However, in this manuscript, it has a point which needs to be further explained and improved:

1. The parameters of algorithms and formulas should be explained more clearly, for example, the definition of symbols, to enhance the readability of the article.

Author Response

Response to Referee#3 ______________________________

Referee#3’s major concerns about this paper.

This paper gives a clear description of the issue to be addressed, and which is an interesting topic indeed. However, in this manuscript, it has a point which needs to be further explained and improved: The parameters of algorithms and formulas should be explained more clearly, for example, the definition of symbols, to enhance the readability of the article.

RESPONSE: Thanks for your comments. Some preliminary symbols have been defined in Section 2. To make the algorithms in the paper clearer, we have added new comments in Algorithms 1, 4, and 5. We have carefully proof-read the paper and believe that all symbols and formulas have been defined and explained.

Round 2

Reviewer 1 Report

Authors have clarified the novelty of the paper and all my issues in the previous version have been addressed. The paper is now ready for publishing. 

Author Response

The reviewer has already been satisfied with the revision.